# July effect in hospitalized cirrhosis patients: A US nationwide study using difference-in-differences analysis

**Melis Gokce Celdir**[1], **George Wehby**[2,3], **Shahana Prakash**[4], **Tomohiro Tanaka**[1] *

**1** Division of Gastroenterology and Hepatology, University of Iowa, Iowa City, IA, United States of America,
**2** Department of Health Management and Policy, University of Iowa College of Public Health, Iowa City, IA,
United States of America, **3** National Bureau of Economic Research, Cambridge, Massachusetts, United
States of America, **4** Department of Internal Medicine, University of Iowa, Iowa City, IA, United States of
America

* tomohiro-tanaka@uiowa.edu

STATES OF AMERICA

**Data Availability Statement:** The data that support
the findings of this study are openly available in the
Healthcare Cost and Utilization Project website
(https://hcup-us.ahrq.gov/).

## Abstract

### Background

The July effect in US teaching hospitals has been studied with conflicting results. We aimed
to evaluate the effect of physician turnover in July on the clinical outcomes of patients hospi-
talized with cirrhosis.

### Methods

We utilized the Nationwide Inpatient Sample database (2016–2019) to identify patients hos-
pitalized with cirrhosis and liver-related complications (variceal bleeding, hepatorenal syn-
drome, acute-on-chronic liver failure). We used difference-in-differences analysis to
compare teaching and non-teaching hospital differences in mortality and length of stay
(LOS) in May and July, and trends in outcomes in other months before and after July.

### Results

We included 78,371 hospitalizations in teaching and 23,518 in non-teaching hospitals in
May and July. Teaching hospital admissions had overall higher complication rates and mor-
tality compared to non-teaching hospitals. We did not find a difference in mortality between
teaching and non-teaching hospitals in all cirrhotic patients (adjusted odds ratio 1.01, 95%
CI [0.88–1.16]) or in those with severe complications (0.87, [0.72–1.06]). There was greater
LOS in July vs. May in teaching hospitals relative to non-teaching hospitals for all patients
with cirrhosis (adjusted rate ratio 1.03, 95%CI [1.02–1.05]) and for those with severe compli-
cations (1.19, [1.17–1.21]). The months after July were associated with longer LOS in teach-
ing hospitals, with the effect gradually diminishing over the subsequent months.

### Conclusions

Our study suggests trainee turnover in July did not affect mortality, but lengthened hospi-
tal stays for patients with cirrhosis, highlighting the need for effective supervision of new

**Funding:** The author(s) received no specific funding for this work.

**Competing interests:** The authors have declared that no competing interests exist.

**Abbreviations:** ACLF, acute-on-chronic liver failure; aRR, adjusted rate ratio; aOR, adjusted odds ratio; CirCom, Cirrhosis Comorbidity Score; HCUP, Healthcare Cost and Utilization Project; LOS, length of stay; NACSLED-ACLF, North American Consortium for the Study of End-Stage Liver Disease Acute on Chronic Liver Failure; NIS, National Inpatient Sample; OR, odds ratio; RR, rate ratio.

trainees and strategies to mitigate operational disruptions for improved clinical management.

## Introduction

The anticipated departure of experienced trainee physicians and entry of a new cohort of trainees in the transition time in July has raised questions about potential disruptions in workflow due to the exit and entry of a sizeable proportion of staff within a short period and the consequent effects on productivity and patient outcomes. The influx of less experienced physicians in July may disrupt workflow, potentially leading to adverse outcomes when new residents and fellows begin training in US teaching hospitals. This potential "July effect" has been studied for surgical outcomes, in-hospital mortality, efficiency measured by length of stay (LOS) and hospital charges, and medication errors, with conflicting results.[1–3] Some studies have indicated small but significant increases in mortality during this period, especially for patients requiring more complex care, such as patients with high-risk myocardial infarction [4] and those undergoing complex cardiac surgeries [5], while other studies have not shown differences in mortality or morbidity outcomes [2, 6–8].

Cirrhosis is the eighth-leading cause of death in the US and imposes a significant healthcare burden with high and increasing rates and costs of hospitalization [9, 10]. Management of hospitalized patients with cirrhosis is complex; early recognition and treatment of complications are necessary, and hospitalizations often require involvement of multidisciplinary care teams and advanced discharge planning [11–13]. Severe in-hospital complications, including acute-on-chronic liver failure (ACLF) occur in 25% of hospitalized patients with decompensated cirrhosis and are associated with high in-hospital mortality [14, 15]. In decompensated hospitalized patients with severe complications and ACLF, identification and management of complications, and prognostication for liver transplantation or initiation of palliative care are complex and challenging. Furthermore, care pathways and protocols targeting hospitalized cirrhosis patients are not well-established [13, 16–18]. Due to trainee turnover and training and the adjustment time of new trainees starting in July, time-sensitive clinical decisions when caring for this complex patient population may be delayed at the beginning of the academic year, potentially leading to adverse clinical outcomes, including mortality, health complications, and longer LOS. However, the impact of trainee physician turnover on the clinical outcomes of hospitalized patients with cirrhosis occurring in July has not been sufficiently examined in the existing literature.

We aimed to evaluate the effect of physician turnover in July on the clinical outcomes of a representative sample of hospitalized patients with cirrhosis in the US using the National Inpatient Sample dataset, employing a quasi-experimental difference-in-differences design. We hypothesized that mortality and LOS may increase in teaching hospitals in July due to trainee physician turnover.

## Material and methods

### Data source and study sample

Our study used data from the National Inpatient Sample (NIS) of the Healthcare Cost and Utilization Project (HCUP) collected by the Agency for Healthcare Research and Quality (https://hcup-us.ahrq.gov/nisoverview.jsp) [19]. The NIS represents approximately 20% of all discharges from community hospitals in the US in a given year. We included years 2016 through

2019 because ICD-10 codes were first used in the fourth quartile of 2015 and because admissions and outcomes in 2020 were affected by the COVID-19 pandemic. In the NIS database, a hospital is categorized as a teaching hospital if it has ACGME-approved residency programs, is a member of the Council of Teaching Hospitals, or maintains a resident-to-bed ratio exceeding 0.25. Due to the low number of rural hospitals in the dataset and the absence of information regarding their teaching or non-teaching status (i.e., facilities are coded as urban-teaching, urban-non-teaching, or rural), we excluded rural hospitals from our analyses. We obtained information on the following additional hospital-level characteristics from the NIS database: hospital bed size (small, medium, large), nine census divisions, and ownership status (public, private not-for-profit, private investor-owned).

## Identification of comorbidities and outcomes

We identified hospitalized cirrhosis patients using ICD-10 codes as listed in the S1 Table. The accuracy of these ICD-10 diagnostic codes in detecting cirrhosis admissions has been demonstrated in previous work [20]. In sub analyses, we focused on patients at higher risk of inpatient mortality who have severe complications during their hospital stay, including acute-on-chronic liver failure, hepatorenal syndrome, or variceal bleeding. Acute-on-chronic liver failure (ACLF) was identified based on the North American Consortium for the Study of End-Stage Liver Disease Acute on Chronic Liver Failure (NACSLED-ACLF) score, using two or more organ failures in patients with chronic liver failure [21]. NACSLED-ACLF defined organ failures are respiratory failure (need for mechanical ventilation), renal failure (acute kidney injury or need for dialysis), cardiovascular failure (shock, central venous pressure procedure, arterial line procedure), and neurological failure (brain death or hepatic coma) [22]. The ICD-10 codes that we used to identify the presence of ACLF, hepatorenal syndrome, and variceal bleeding are provided in S1 Table. We also identified relevant diagnoses to calculate Cirrhosis Comorbidity Score (CirCom) [23]. CirCom is a validated comorbidity scoring system for patients with cirrhosis developed for observational studies based on the following comorbidities: chronic obstructive pulmonary disease, acute myocardial infarction, peripheral arterial disease, epilepsy, substance abuse, heart failure, cancer, and chronic kidney disease [23–25].

## Statistical analyses

We first provided a description of the patients, encompassing their demographic and clinical characteristics. Continuous variables were compared using the Student's t test, while categorical variables were analyzed using the chi-square test.

We then estimated a difference-in-differences model that compared the change in in-hospital mortality and LOS of hospitalized patients with cirrhosis during May vs. July and between teaching and non-teaching hospitals. Specifically, the difference in outcomes for May vs. July in teaching hospitals was compared with the difference in outcomes for the month of May and the month of July in non-teaching hospitals. This model assumes that any seasonal variation in outcomes between these two months that is not related to resident turnover (in teaching hospitals) is otherwise similar between teaching and non-teaching hospitals. We used May as the pre-transition month rather than June because some residency programs start training within June, or residents may leave early at the end of their training. We estimate the difference-in-differences regression using logistic regression models for mortality. Because LOS was not normally distributed and positively skewed, we used Poisson regression where LOS was seen as a count of number of days that the patient remained in the hospital [26, 27]. The models included an interaction term, (*July*)×(*Teaching*), where *July* is a binary indicator for outcomes in July relative to May, and Teaching is a binary indicator for hospital teaching versus non-

teaching status, as well as these indicators as covariates on their own. We first estimated a basic difference-in-differences specification without additional covariates. We also estimated a specification that adjusts for hospital ownership, census division, bed size, and patient's sex, age, race, and Cirrhosis Comorbidity Score (CirCom). We estimated the models for the total sample of patients with cirrhosis and separately for those with severe complications as defined above. We report the estimates as odds ratios (OR) for in-hospital mortality and incidence rate ratios (RR) for length of stay, which are the exponentiated coefficients of the difference-in-differences interaction term. The RR for length of stay specifically compares the counted days of hospitalization between groups. For example, an RR of 2 in one group, relative to another, implies that the counted days of hospitalization (i.e., LOS) are twice as long in the former group.

To better understand seasonal variation in the months leading to July between teaching and non-teaching hospitals, as well as variation in outcomes post-July, we estimated an extended version of the difference-in-differences model that includes all other calendar months. This second model included separate 0/1 indicators for each month (except May, which was the reference month) and interactions with teaching status similar to the first model [28]. The exponentiated coefficients of the difference-in-differences interaction terms were plotted for each month (with May as the reference month). The interaction term in the form of (Month)* (Teaching) with an indicator variable for each month and hospital teaching status from the fitted adjusted logistic regression models for mortality and Poisson models for length of stay is presented in the Figs 1–4. The exponentiated coefficients were plotted according to the month. Error bars indicate 95% confidence intervals in the figures.

We used R version 4.3.0 for statistical analyses and considered p values <0.05 statistically significant.

## Ethical statements

This cross-sectional nationwide study conforms to the ethical guidelines 1975 Declaration of Helsinki and its later amendments or comparable ethical standards. The Human Investigation Committee (IRB) of the University of Iowa was not required to approve or review this study as the National Inpatient Sample provides publicly available and de-identified data.

## Results

### Sample description

The analytical sample included 78,371 hospitalizations in teaching hospitals in May (N = 39283) and July (N = 39088) and 23,518 in non-teaching hospitals in May (N = 11837) and July (N = 11681). Patient characteristics are summarized in Table 1. Patients were slightly younger, on average, in teaching hospitals compared with non-teaching hospitals (58.4 vs. 59.2 years in teaching vs. non-teaching, respectively, p < 0.01). The proportion of women was also slightly lower in teaching hospitals (38.2% in teaching vs. 39.2% in non-teaching, p<0.01). Inpatient mortality for cirrhosis admission combining the two months was 5.7% in teaching vs. 5.1% in non-teaching hospitals. The distribution of CirCom scores was similar between teaching and non-teaching hospitals. Rates of inpatient cirrhosis complications were higher in teaching hospitals.

### Difference-in-differences estimates, May versus July

Table 2 summarizes the results of difference-in-differences estimates for models for mortality and LOS mentioned in this section. Unadjusted inpatient mortality in the total sample was not

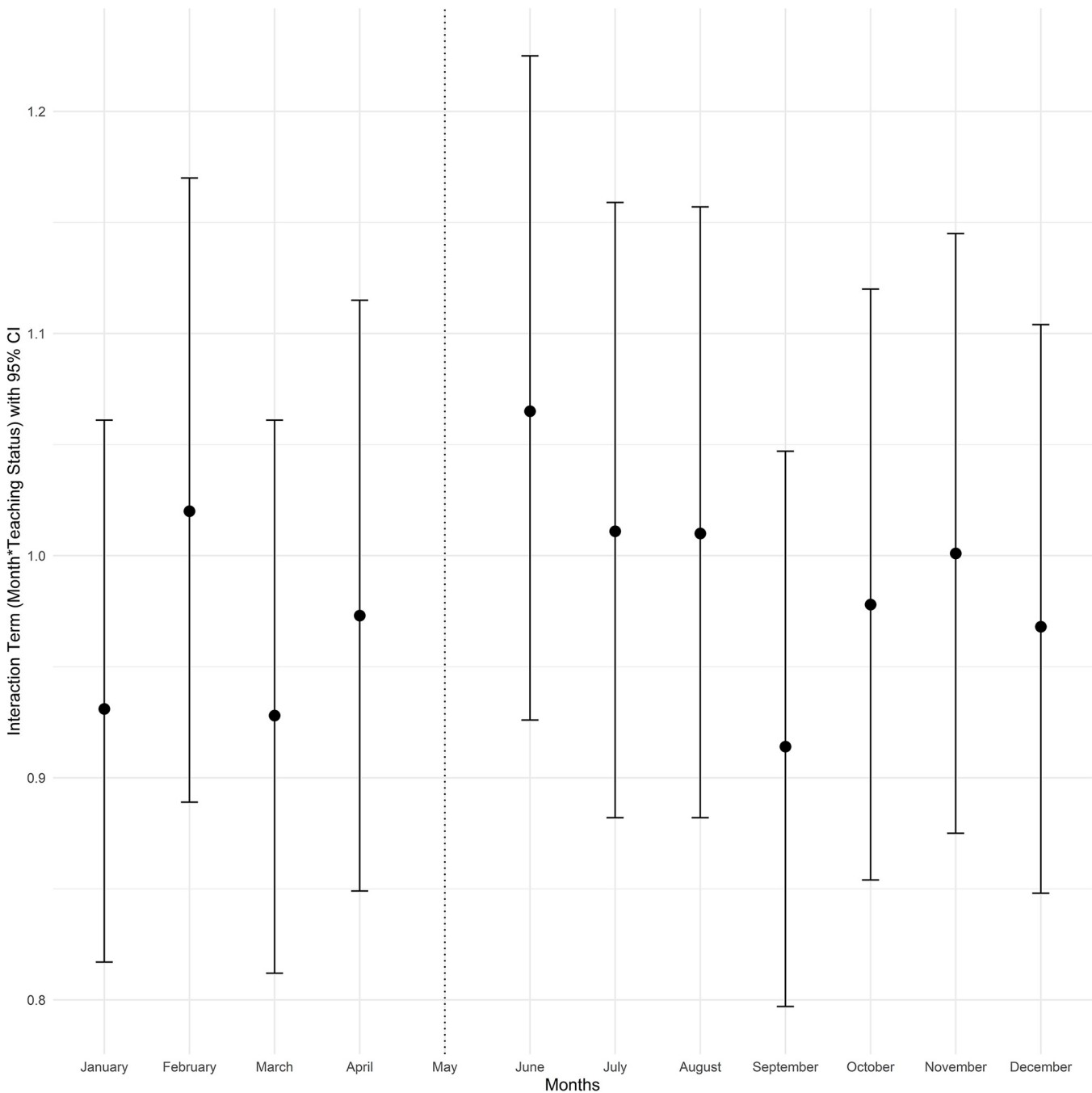

**Fig 1. Month and teaching status interaction term according to the calendar month for mortality, in all cirrhosis admissions.**

different in teaching hospitals between May and July (4.9% versus 4.7%) or in non–teaching hospitals between May and July (5.6% versus 5.5%). The unadjusted difference-in-differences estimate comparing July and May mortality differences between teaching and non-teaching hospitals was not statistically significant (OR = 1.01 95% CI 0.89–1.16). Adjusting for hospital and patient characteristics resulted in a null difference-in-differences estimate (OR 1.01, 95% CI 0.89–1.15). Similarly, among patients with severe complications, the difference-in-

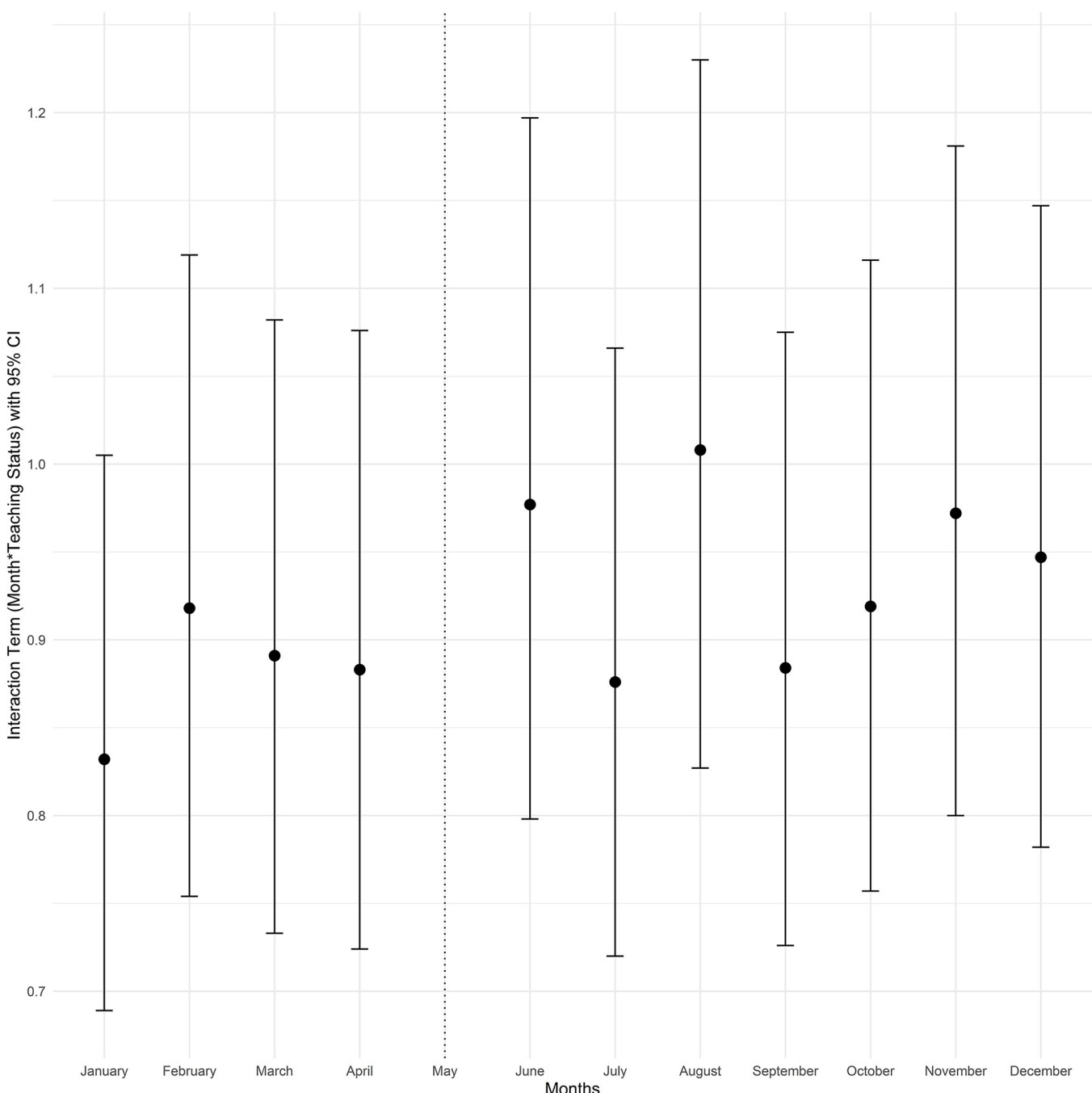

**Fig 2. Month and teaching status interaction term according to the calendar month for mortality in cirrhosis admissions with severe complications.**

differences comparisons showed no significant differences (OR = 0.87; 95% CI of 0.72–1.06 in the adjusted model).

LOS (i.e., the counted days of hospitalization) in non-teaching hospitals was 3% shorter in July than in May (adjusted rate ratio 0.97, 95% CI 0.97–0.98). In contrast, it was similar in teaching hospitals between May and July (adjusted rate ratio 0.99, 95% CI 0.98–1.00). The difference-in-differences analysis of LOS revealed a more extended hospitalization in July versus May in teaching hospitals compared to non-teaching hospitals (unadjusted and adjusted rate

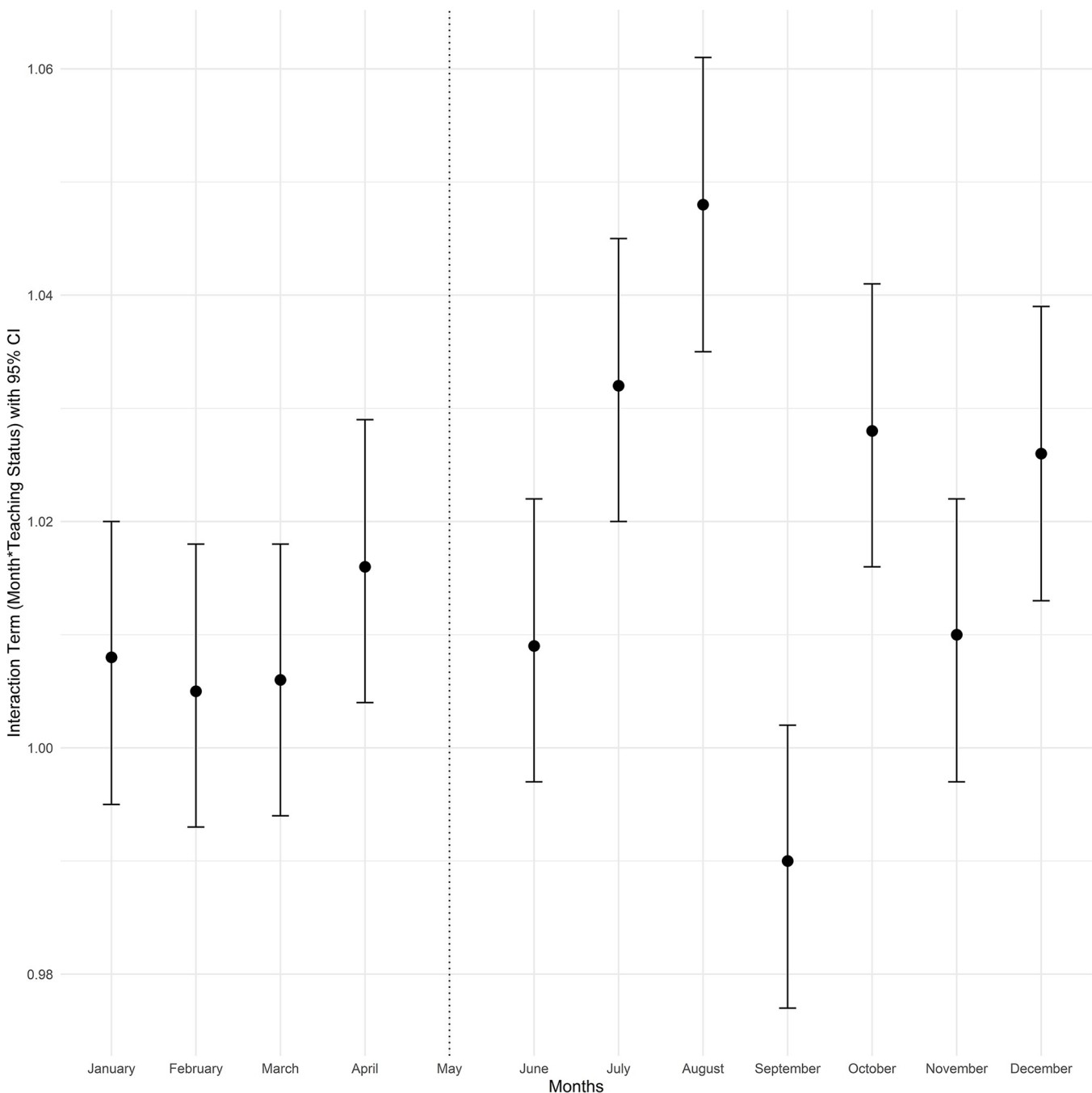

**Fig 3. Month and teaching status interaction term according to the calendar month for length of stay in all cirrhosis admissions.**

ratio 1.03, 95% CI 1.02–1.05), interpreted as 3% increase in LOS in teaching hospitals attributed to the effect of July, compared to non-teaching hospitals. Similarly, among patients with severe complications, non-teaching hospitals had shorter LOS in July than in May (adjusted rate ratio 0.93, 95% CI 0.91–0.96), while teaching hospitals had similar LOS between July and May (adjusted rate ratio 1.01, 95% CI 0.99–1.02). The difference-in-differences estimates indicated a longer hospitalization in July versus May in teaching hospitals (adjusted rate ratio 1.19,

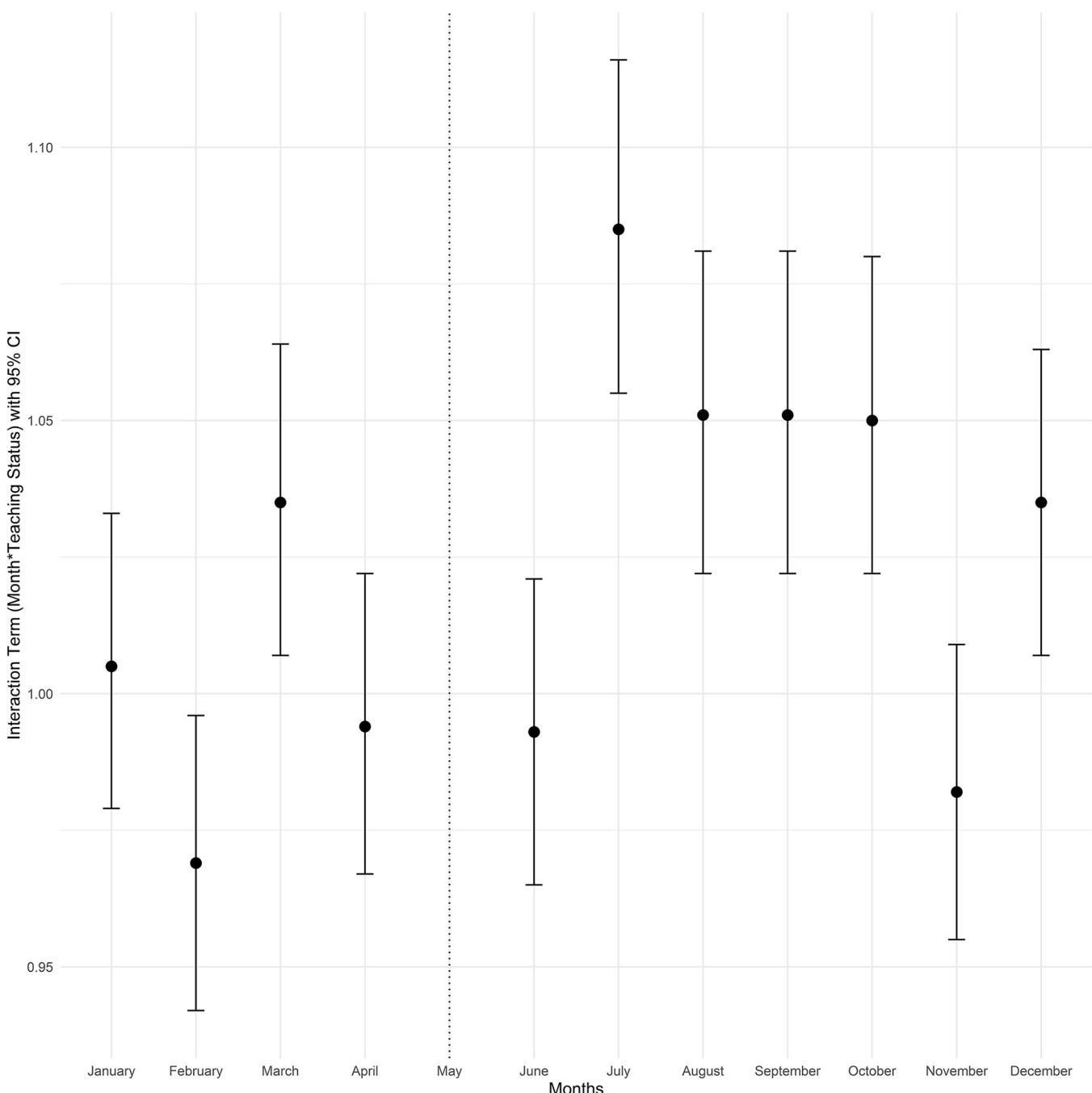

**Fig 4. Month and teaching status interaction term according to the calendar month for length of stay cirrhosis admissions with severe complications.**

95% CI 1.17–1.21), interpreted as 19% increase in LOS in teaching hospitals attributed to the effect of July.

## Difference-in-differences estimates, month-by-month

In both teaching and non-teaching hospitals, mortality in January, February, November, and December was higher than in the rest of the calendar year. Teaching hospitals had higher

**Table 1. Demographic and clinical characteristics of all hospitalized cirrhosis patients, according to teaching hospital status (2016–2019).**

| | Teaching hospitals | | | Non-teaching hospitals | | |
|---|---|---|---|---|---|---|
| | May (N = 39283) | July (N = 39088) | p-value | May (N = 11837) | July (N = 11681) | p-value |
| Female, n (%) | 14998 (38.2) | 14925 (38.2) | 0.99 | 4639 (39.2) | 4565 (39.1) | 0.88 |
| Mean patient age, y (SD) | 58.5 (13.1) | 58.4 (13.2) | 0.04 | 59.3 (13.3) | 59.0 (13.2) | 0.06 |
| Race (%) | | | 0.33 | | | 0.35 |
| White | 24437 (63.9) | 24441 (64.1) | | 8001 (68.9) | 8002 (69.9) | |
| Black | 4591 (12.0) | 4559 (12.0) | | 861 (7.4) | 859 (7.5) | |
| Hispanic | 6525 (17.1) | 6577 (17.3) | | 2065 (17.8) | 1934 (16.9) | |
| Other or unknown | 2698 (7.1) | 2545 (6.7) | | 910 (7.7) | 1066 (9.1) | |
| Cirrhosis Comorbidity Index | | | 0.25 | | | 0.20 |
| 0 + 0 | 14901 (37.9) | 14578 (37.3) | | 4321 (36.5) | 4362 (37.3) | |
| 1 + 0 | 10162 (25.9) | 10349 (26.5) | | 3211 (27.1) | 3205 (27.4) | |
| 1 + 1 | 2902 (7.4) | 2927 (7.5) | | 1046 (8.8) | 983 (8.4) | |
| 3 + 0 | 9847 (25.1) | 9798 (25.1) | | 2892 (24.4) | 2739 (23.4) | |
| 3 + 1 | 652 (1.7) | 602 (1.5) | | 175 (1.5) | 178 (1.5) | |
| 5 + 0 | 185 (0.5) | 203 (0.5) | | 39 (0.3) | 32 (0.3) | |
| 5 + 1 | 634 (1.6) | 631 (1.6) | | 153 (1.3) | 182 (1.6) | |
| NACSLED >1, n (%) | 2802 (7.1) | 2631 (6.7) | 0.03 | 667 (5.6) | 619 (5.3) | 0.27 |
| Cardiac failure, n (%) | 2092 (5.3) | 1980 (5.1) | 0.10 | 519 (4.4) | 526 (4.5) | 0.68 |
| Cerebral failure, n (%) | 204 (0.5) | 165 (0.4) | 0.05 | 67 (0.6) | 54 (0.5) | 0.31 |
| Respiratory failure, n (%) | 2862 (7.3) | 2740 (7.0) | 0.13 | 725 (6.1) | 640 (5.5) | 0.04 |
| Renal failure, n (%) | 11139 (28.4) | 11180 (28.6) | 0.45 | 2943 (24.9) | 2958 (25.3) | 0.42 |
| Hepatorenal syndrome, n (%) | 1721 (4.4) | 1702 (4.4) | 0.87 | 422 (3.6) | 401 (3.4) | 0.61 |
| Variceal bleeding, n (%) | 2827 (7.2) | 2595 (6.6) | <0.05 | 760 (6.4) | 743 (6.4) | 0.87 |

unadjusted mortality rates than non-teaching hospitals throughout the year (S2 Table). Median LOS was 4 days in the overall patient population (IQR: 2–7, S3 Table). Consistent with the analyses of patients hospitalized only in May and July, there was no significant trend of mortality differences based on teaching status and before or after July. However, July and the following months were associated with longer LOS in teaching hospitals, and this difference gradually tapered over the subsequent months. The month-by-month (relative to May) difference-in-differences estimates for mortality and LOS are plotted in Figs 1–4.

## Discussion

In a national inpatient sample of patients hospitalized with cirrhosis, we found no evidence of an effect of physician turnover in July on mortality in teaching hospitals using a difference-in-differences analysis comparing teaching and non-teaching hospitals in May vs. July. We also found no evidence of an effect when focusing on patients with severe complications and no differences between teaching and non-teaching hospitals in seasonal trends in mortality in other months leading to or following July. Our findings revealed a modest yet statistically significant increase in LOS at teaching hospitals in July, compared to non-teaching hospitals, determined using a difference-in-differences design. Moreover, this effect was more pronounced in patients with severe complications, where there was a 19% increase in LOS attributed to the 'July effect' in teaching hospitals as opposed to non-teaching hospitals.

To our knowledge, this study is the first to assess the effect of trainee physician turnover in July in teaching hospitals on the health outcomes of hospitalized patients with cirrhosis.

**Table 2. Adjusted inpatient mortality and length of stay in cirrhosis admissions in May and July, according to teaching status and comparison to non-teaching hospitals.**

| | Mortality in the total cohort | | | | Mortality among patients with severe complications | | | |
|---|---|---|---|---|---|---|---|---|
| | May | July | aOR[1] (95% CI) | aOR of the May-July Difference in Teaching Compared with Non-Teaching[2] (95% CI) | May | July | aOR[1] (95% CI) | aOR of the May-July Difference in Teaching Compared with Non-Teaching[2] (95% CI) |
| **Non-Teaching, %** | 4.9% | 4.7% | 0.98 (0.87, 1.11) | - | 18.8% | 20.8% | 1.18 (0.99, 1.41) | - |
| **Teaching, %** | 5.6% | 5.5% | 0.98 (0.92, 1.05) | 1.01 (0.88–1.15) | 21.5% | 21.8% | 1.02 (0.93, 1.11) | 0.87 (0.72–1.06) |
| | Length of stay in total cohort | | | | Length of stay among patients with severe complications | | | |
| | May | July | aRR[3] (95% CI) | aRR of the May-July Difference in Teaching Compared with Non-Teaching[4] (95% CI) | May | July | aRR[3] (95% CI) | aRR of the May-July Difference in Teaching Compared with Non-Teaching[4] (95% CI) |
| **Non-Teaching, days, median [IQR]** | 4 [2, 7] | 4 [2, 6] | 0.97 (0.95, 0.98) | - | 5 [3, 10] | 5 [3, 9] | 0.93 (0.91, 0.96) | - |
| **Teaching, days, median [IQR]** | 4 [2, 7] | 4 [2, 7] | 0.99 (0.98, 1.00) | 1.03 (1.02, 1.05) | 6 [3, 11] | 6 [3, 12] | 1.01 (0.99, 1.02) | 1.19 (1.17, 1.21) |

[1]Comparison of mortality in July relative to May according to hospital type

[2]Comparison of July-May mortality odds ratio for patients in teaching hospitals relative to the odds ratio for patients hospitalized in non-teaching hospitals

[3]Rate difference based on the number of days of length of stay in July compared to May according to hospital type

[4]Comparison of July-May rate ratio of length of stay for patients in teaching hospitals relative to the rate ratio for patients hospitalized in non-teaching hospitals. Models are adjusted for sex, age, race, hospital ownership, census division, bed size, Cirrhosis Comorbidity Score. The exponentiated coefficients were plotted according to the month. aOR: Adjusted odds ratio. aRR: Adjusted rate ratio.

Contrary to our results, which revealed no observable impact of July on mortality rates among hospitalized patients with cirrhosis, a systematic review concluded that there is an association between July turnover and mortality, with an effect size ranging from an overall relative risk increase of 4.3 to 12%. However, there is significant heterogeneity in the existing literature [2]. The elevation in mortality risk was more consistently observed in studies on patient groups with elevated risk, such as complex cardiac surgeries [5]. Among medical subspecialties, studies were performed primarily for cardiac outcomes and did not show mortality differences [4, 8, 29]; except for a small but significant increase for higher-risk cardiac patients [4].

LOS is commonly used as a proxy for efficient clinical management and resource utilization in healthcare settings [30]. Our results are consistent with several prior studies that showed increased LOS during the trainee turnover period [1, 5, 31, 32]. However, most of these studies did not consider seasonal variation using a difference-in-differences framework. Noteworthy of our findings include the difference-in-differences estimate suggesting an increase in LOS in teaching hospital. However, this increase might be arguably attributable to a small decline in LOS in non-teaching hospitals between May and July, while LOS remained steady in teaching hospitals. Therefore, this finding on LOS should be cautiously interpreted. The difference-in-differences model and this finding hinge on the assumption that the May-July difference would have been similar between teaching and non-teaching hospitals without the trainee turnover, which cannot be directly tested. Therefore, we have also performed a sensitivity analysis on trends of LOS before and after July. The month-by-month analysis leading to July shows no evidence of pre-July trends that would explain the estimated increase in July but there was a consistent trend of increased LOS in month-by-month RRs in the months

following July in teaching hospitals compared to non-teaching hospitals. Additionally, considering the unclear reasons for non-teaching hospitals having shorter LOS in July compared to May, it seems reasonable to hypothesize that common confounding factors, which potentially reduce the LOS in July, offset the "July-effect" due to trainee turnover in teaching hospitals, which could have led to a bias towards a null effect. The lower level of familiarity of new trainees with the teams, care coordination of cirrhosis patients with complex discharge planning needs [13, 16, 17], and delays in initiating palliative care [16] may have resulted in longer hospital stays in July and the following months. Establishing care pathway protocols may mitigate these factors, as it has been shown to improve outcomes in chronic diseases such as heart failure [33].

Our study also offers new evidence on seasonal variation in two important health outcomes for patients with cirrhosis. Specifically, our data suggest increased mortality and LOS during winter months, which is consistent with studies demonstrating seasonal variations in mortality and hospitalization outcomes in general medicine and other specialties [34, 35].

Our study has two notable strengths: the inclusion of data from a national inpatient sample and the use of difference-in-differences analysis with non-teaching hospitals as a control group to account for seasonal variation. Nevertheless, our study has several limitations. First, although we have adjusted for multiple potential patient and hospital level confounders, there may still be confounding between teaching and non-teaching hospitals over time that bias the difference-in-differences estimates. Second, there are potential misclassification errors. For example, we used the Cirrhosis Comorbidity Score to account for patients' pre-existing morbidity; however, this scoring has been only validated in the outpatient setting and may not apply to our inpatient sample [23, 25, 36]. Also, we have relied on administrative diagnostic codes, and there may be differences in coding practices between teaching and non-teaching hospitals. Inherent to administrative databases, NIS ICD-10 codes may not always be accurate for cirrhosis and related complications [20]. Additionally, the NIS dataset does not provide laboratory or medication data. Third, severe complications defined in our dataset did not differentiate the cirrhosis complications present at admission from those developed during hospitalization. Due to the limitations of registry-based data, we did not have laboratory and further clinical data to stratify disease severity at the time of admission or further adjust for severity in the models. Fourth, each year, the NIS sample may include different hospitals, which could differentially alter the sample composition over time between teaching and non-teaching status. We adjusted for several patient and hospital-level covariates, which accounts for any sample composition changes in these specific characteristics and found similar results overall. Finally, our NIS dataset entries are based on admissions, not individuals; we were not able to consider multiple admissions for a patient and the difference in readmissions between teaching and non-teaching hospitals in our analyses.

As a potential future direction, examining additional outcomes, including readmission rates and post-discharge mortality, would be valuable. However, these outcomes cannot be studied using this specific dataset. The readmission rate for patients with cirrhosis-related complications has been reported to be approximately 25% [37]. Previous studies report that non-teaching hospitals have higher readmission rates, [37] and premature discharge may be associated with higher readmissions [38]. Furthermore, the trend of increasing 30-day post-discharge mortality despite a decrease in inpatient mortality in recent years [10], signifies a shift of mortality risks to after discharge. This underscores the importance of careful discharge planning to balance efforts to reduce LOS with the prevention of premature discharge [39].

## Conclusion

We did not find any differences in in-hospital mortality in July in teaching hospitals for patients with cirrhosis, but there was an increase in LOS in July in teaching relative to non-teaching hospitals, especially in cirrhosis patients with severe complications. The findings support the need for appropriate supervision of new trainees, developing quality measures to minimize operational disruptions, and establishing care protocols for hospitalized cirrhosis patients, which may improve clinical outcomes in this high-risk patient population.

## Supporting information

**S1 Table. ICD-10 and CPT codes to identify the study population and severe cirrhosis complications.**
(DOCX)

**S2 Table. Inpatient mortality among patients admitted to teaching and non-teaching hospitals, according to month.**
(DOCX)

**S3 Table. LOS among patients admitted to teaching and non-teaching hospitals, according to month.**
(DOCX)

## Acknowledgments

**Presentation.** A portion of the material presented in this article was also featured in a presentation at the AASLD Liver Meeting in November 2023.

## Author Contributions

**Conceptualization:** Melis Gokce Celdir, George Wehby, Tomohiro Tanaka.

**Data curation:** Melis Gokce Celdir, Tomohiro Tanaka.

**Formal analysis:** Melis Gokce Celdir, Tomohiro Tanaka.

**Methodology:** George Wehby, Tomohiro Tanaka.

**Software:** Shahana Prakash, Tomohiro Tanaka.

**Supervision:** George Wehby, Tomohiro Tanaka.

**Writing – original draft:** Melis Gokce Celdir.

**Writing – review & editing:** George Wehby, Shahana Prakash, Tomohiro Tanaka.

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
