## [Decision Letter · Decision Letter 0]

20 Oct 2024

PONE-D-24-30541July effect in hospitalized cirrhosis patients: a US nationwide study using difference-in-differences analysisPLOS ONE

Dear Dr. Celdir,

Thank you for submitting your manuscript to PLOS ONE. After careful consideration, we feel that it has merit but does not fully meet PLOS ONE’s publication criteria as it currently stands. Therefore, we invite you to submit a revised version of the manuscript that addresses the points raised during the review process.

We look forward to receiving your revised manuscript.

Kind regards,

Usama Waqar, M.B.B.S

Academic Editor

PLOS ONE

Additional Editor Comments (if provided):

Reviewers' comments:

Reviewer's Responses to Questions

**Comments to the Author**

1. Is the manuscript technically sound, and do the data support the conclusions?

Reviewer #1: Yes

Reviewer #2: Yes

2. Has the statistical analysis been performed appropriately and rigorously? 

Reviewer #1: Yes

Reviewer #2: Yes

3. Have the authors made all data underlying the findings in their manuscript fully available?

Reviewer #1: Yes

Reviewer #2: Yes

4. Is the manuscript presented in an intelligible fashion and written in standard English?

Reviewer #1: Yes

Reviewer #2: Yes

5. Review Comments to the Author

Reviewer #1: Overall a good study given appears to be novel in looking at the impact of the July effect. It is slightly surprising that there was not significant differences in mortality at all and especially major complications in the teaching hospitals with the new trainees. It would be interesting to see if there would be a difference in terms of types of trainees as not all teaching hospitals have GI fellowships. So if the teaching hospitals include programs without GI fellowships, would the outcomes be different as many patients get admitted to community teaching hospitals that have residency programs but no GI fellowships.

Reviewer #2: Summary: This study investigates the "July effect" by examining the impact of trainee physician turnover on clinical outcomes in cirrhosis patients admitted to US teaching hospitals. Conflicting evidence exists around the July effect, however, this study utilizes a difference-in-differences analysis, comparing outcomes in May and July for teaching versus non-teaching hospitals. The key outcomes measured are mortality and length of stay (LOS), with the results showing no significant difference in mortality but a modest increase in LOS in teaching hospitals, particularly for patients with severe complications. The study emphasizes the need for better supervision of new trainees to mitigate operational disruptions.

Overall Impression: The manuscript is well-written and easy to follow, with clear articulation of the research question, methods, and results. It is methodologically strong and addresses an important research question, contributing valuable insights into the ongoing debate on the July effect. The use of a large, representative dataset (NIS) and a rigorous statistical approach (difference-in-differences) strengthens the validity of the findings. However, there are several areas where further clarification are required, as discussed below.

1. Potential confounding factors: July effect might interact with factors not captured in the dataset, such as differences in hospital staffing structures or the availability of specialized care for cirrhosis patients

6. PLOS authors have the option to publish the peer review history of their article (what does this mean?). If published, this will include your full peer review and any attached files.

Reviewer #1: No

Reviewer #2: No

---

## [Author Response · Author response to Decision Letter 0]

16 Nov 2024

Response to Reviewers 

Reviewer #1: Overall a good study given appears to be novel in looking at the impact of the July effect. It is slightly surprising that there was not significant differences in mortality at all and especially major complications in the teaching hospitals with the new trainees. It would be interesting to see if there would be a difference in terms of types of trainees as not all teaching hospitals have GI fellowships. So if the teaching hospitals include programs without GI fellowships, would the outcomes be different as many patients get admitted to community teaching hospitals that have residency programs but no GI fellowships.

Response: Thank you; this is an excellent point. Unfortunately, the NIS database does not contain data specifically on the presence of GI fellowships. NIS obtains the hospital’s teaching status from the AHA Annual Survey of Hospitals, and a hospital is designated if it has at least one ACGME-approved residency or a ratio of full-time equivalent interns and residents to beds of 0.25 or higher. GI fellowship-specific information can provide further insight into whether subspecialty consultation with trainee involvement affects outcomes in months following new trainee recruitment. We have added this limitation and discussion point to the manuscript. (Line 287 in the clean version of the manuscript body)

Reviewer #2: Summary: This study investigates the "July effect" by examining the impact of trainee physician turnover on clinical outcomes in cirrhosis patients admitted to US teaching hospitals. Conflicting evidence exists around the July effect, however, this study utilizes a difference-in-differences analysis, comparing outcomes in May and July for teaching versus non-teaching hospitals. The key outcomes measured are mortality and length of stay (LOS), with the results showing no significant difference in mortality but a modest increase in LOS in teaching hospitals, particularly for patients with severe complications. The study emphasizes the need for better supervision of new trainees to mitigate operational disruptions.

Overall Impression: The manuscript is well-written and easy to follow, with clear articulation of the research question, methods, and results. It is methodologically strong and addresses an important research question, contributing valuable insights into the ongoing debate on the July effect. The use of a large, representative dataset (NIS) and a rigorous statistical approach (difference-in-differences) strengthens the validity of the findings. However, there are several areas where further clarification are required, as discussed below.

Response: Thank you for the encouraging comments.

1. Potential confounding factors: July effect might interact with factors not captured in the dataset, such as differences in hospital staffing structures or the availability of specialized care for cirrhosis patients

Response: This is an important point that emphasizes the presence of residual confounding even after we adjusted our models and is a limitation of the study's registry-based cross-sectional design. We have now added this limitation in the discussion session. (Line 307 in the clean version of the manuscript body)

---

## [Editor Report · Decision Letter 1]

12 Dec 2024

July effect in hospitalized cirrhosis patients: A US nationwide study using difference-in-differences analysis

PONE-D-24-30541R1

Dear Dr. Celdir,

We’re pleased to inform you that your manuscript has been judged scientifically suitable for publication and will be formally accepted for publication once it meets all outstanding technical requirements.

Kind regards,

Usama Waqar, M.B.B.S

Academic Editor

PLOS ONE

---

## [Editor Report · Acceptance letter]

30 Dec 2024

PONE-D-24-30541R1 

PLOS ONE

Dear Dr. Celdir, 

I'm pleased to inform you that your manuscript has been deemed suitable for publication in PLOS ONE. Congratulations! Your manuscript is now being handed over to our production team.

Kind regards, 

on behalf of

Dr. Usama Waqar 

Academic Editor

PLOS ONE